# High-quality mesoporous graphene particles as high-energy and fast-charging anodes for lithium-ion batteries

Runwei Mo[1], Fan Li[1], Xinyi Tan[1], Pengcheng Xu[1], Ran Tao[1], Gurong Shen[1], Xing Lu[1], Fang Liu[1], Li Shen[1], Bin Xu[2], Qiangfeng Xiao[3], Xiang Wang[4], Chongmin Wang [4], Jinlai Li[5], Ge Wang[6] & Yunfeng Lu[1]

The application of graphene for electrochemical energy storage has received tremendous attention; however, challenges remain in synthesis and other aspects. Here we report the synthesis of high-quality, nitrogen-doped, mesoporous graphene particles through chemical vapor deposition with magnesium-oxide particles as the catalyst and template. Such particles possess excellent structural and electrochemical stability, electronic and ionic conductivity, enabling their use as high-performance anodes with high reversible capacity, outstanding rate performance (e.g., 1,138 mA h g$^{-1}$ at 0.2 C or 440 mA h g$^{-1}$ at 60 C with a mass loading of 1 mg cm$^{-2}$), and excellent cycling stability (e.g., >99% capacity retention for 500 cycles at 2 C with a mass loading of 1 mg cm$^{-2}$). Interestingly, thick electrodes could be fabricated with high areal capacity and current density (e.g., 6.1 mA h cm$^{-2}$ at 0.9 mA cm$^{-2}$), providing an intriguing class of materials for lithium-ion batteries with high energy and power performance.

[1] Chemical and Biomolecular Engineering, University of California, Los Angeles, CA 90095, USA. [2] State Key Laboratory of Supramolecular Structure and Materials, Jilin University, Changchun 130012, China. [3] General Motors Research and Development Center, 30500 Mound Road, Warren, MI 48090, USA. [4] Environmental Molecular Sciences Laboratory, Pacific Northwest National Laboratory, Richland, WA 99352, USA. [5] ENN Group, Langfang, Hebei 065001, China. [6] Beijing Advanced Innovation Center for Materials Genome Engineering, Beijing Key Laboratory of Function Materials for Molecule & Structure Construction, University of Science and Technology Beijing, Beijing 100083, China. Correspondence and requests for materials should be addressed to J. L. (email: lijinlai@vip.126.com) or to G.W. (email: gewang@mater.ustb.edu.cn) or to Y.L. (email: luucla@ucla.edu)

There are increasing demands for lithium-ion batteries (LIBs) with both high-energy density and power density for electric vehicles and mobile device applications[1–3]. Such performance characteristics, in the context of their electrochemical process, are generally governed by the capacity and voltage of the electrodes, as well as the transport of electrons and ions. Current LIBs commonly use graphite anodes, which exhibit a theoretical capacity of 372 mA h g$^{-1}$ and have relatively slow lithium-ion insertion-desertion kinetics, with limited energy density and power performance[4–6]. As an alternative, spinel Li$_4$Ti$_5$O$_{12}$ (LTO) has attracted considerable attention due to its fast and reversible insertion/extraction kinetics for lithium ions[7,8]. Unfortunately, its low theoretical capacity (175 mA h g$^{-1}$) and high redox voltage (1.55 V vs. Li$^+$/Li) limit the energy density. Beyond the above mentioned materials, materials containing high-energy-density components (e.g., silicon with a theoretical capacity of 4200 mA h g$^{-1}$) have also been explored extensively (e.g., graphite–silicon composites)[9]. Such high-energy components, however, often show slow electrochemical kinetics with a significant volume change during the lithiation and delithiation, resulting in unsatisfactory power performance and fast capacity decay[10]. Developing novel materials with high specific capacity, fast ion insertion/extraction kinetics, structural and electrochemical robustness is critical toward LIBs with high-energy-power density and long cycling stability[11–14].

Graphene, which has a theoretical capacity of 744 mA h g$^{-1}$, fast electronic mobility of 10,000 cm$^2$ V$^{-1}$ s and a high lithium-ion diffusivity (10$^{-7}$–10$^{-6}$ cm$^2$ s$^{-1}$)[15–19], holds great promise as an anode material for high-energy-power LIBs. However, such outstanding performance is often achieved in high-quality graphene only. The defects in graphene decrease the electric conductivity, electrochemical and structural stability, which reduce the power density and Coulombic efficiency and shorten the cycling life[20–22]. Moreover, the presence of defects may also cause a slanted charge/discharge profile, which may affect the cell voltage profile and energy density[23,24]. In addition, irreversible stacking of graphene sheets commonly occurs in graphene-based electrodes during the charging and discharging, which reduces the number of lithium-storage sites and ion-diffusion rate, resulting in reduced energy density and shortened cycling life[25–28].

We report herein a synthesis of high-quality mesoporous graphene particles with nitrogen doping as an anode material for high-energy-power LIBs. As depicted in Fig. 1, using mesoporous magnesium oxide (MgO) as the template and catalyst, nitrogen-doped graphene is grown within the MgO particles by chemical vapor deposition (CVD) using acetonitrile as the precursor. Removal of the template leads to the formation of nitrogen-doped mesoporous graphene particles (NMG). It is believed that the growth of graphene on MgO is achieved through by free-radical condensation of hydrocarbons[29]. Compared with metallic catalysts, MgO generally leads to graphene materials with higher defect density[30], which can be reduced by subsequent microwave radiation and enables the synthesis of high-quality, nitrogen-doped, mesoporous graphene (HNMG) particles. The unique architecture of the HNMG particles affords several major advantages leading to high performance. At first, the low defect density of the HNMG particles affords outstanding electronic conductivity and electrochemical stability, improving power performance and cycling stability. Moreover, the three-dimensional (3D) graphene scaffolds provide the HNMG particles with structural stability, avoiding irreversible stacking of the graphene layers during the charging and discharging, which also prolongs the cycling stability and improves the Coulombic efficiency. It is also worth mentioning that the mesoporous structure provides the HNMG particles with a large number of lithium-storage sites, effective ion-transport pathways, as well as space to accommodate the volume change that may occur during charging and discharging. Finally, nitrogen doping improves the electrode–electrolyte interactions, provides lithiophilic surface moieties, and offers uniform nucleation sites with small nucleation overpotential, which has advantages in high-energy density and fast-charging capability.

## Results

**Materials synthesis and characterization**. Figure 2a, b, respectively, present a representative scanning electron microscopy (SEM) and transmission electron microscopy (TEM) image of the MgO template, showing a particular morphology with radially oriented porous channels. The average size of the particles is ~ 7 μm (Supplementary Figure 1). After the CVD process, nitrogen-doped graphene is uniformly deposited on the template particles (Supplementary Figure 2), evidenced by the homogenous dispersion of C, N, O, and Mg in the particles (Fig. 2c, d). Etching

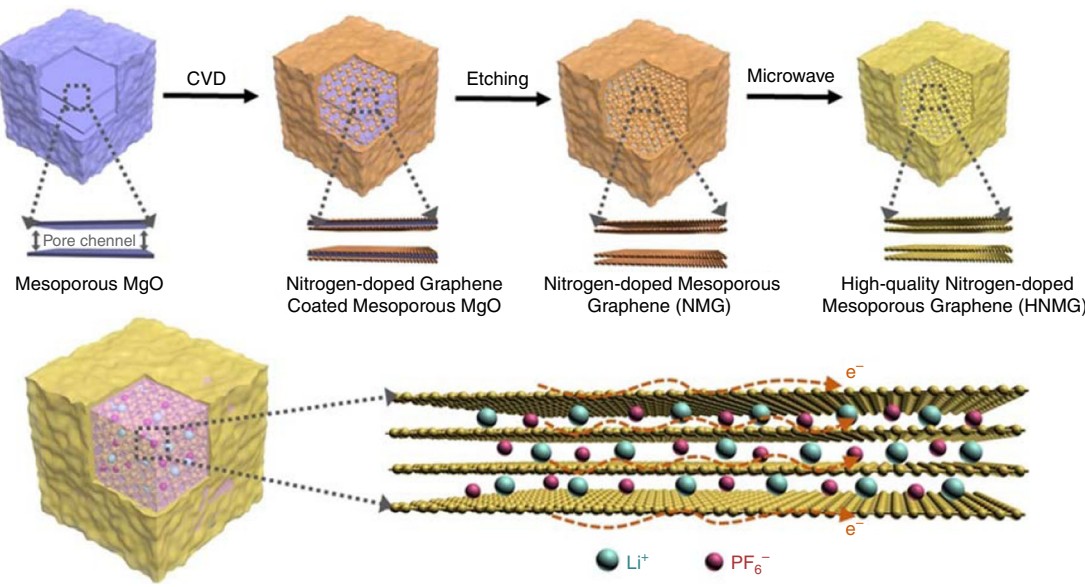

**Fig. 1** A schematic illustrating the synthesis of high-quality, nitrogen-doped, mesoporous graphene (HNMG) particles

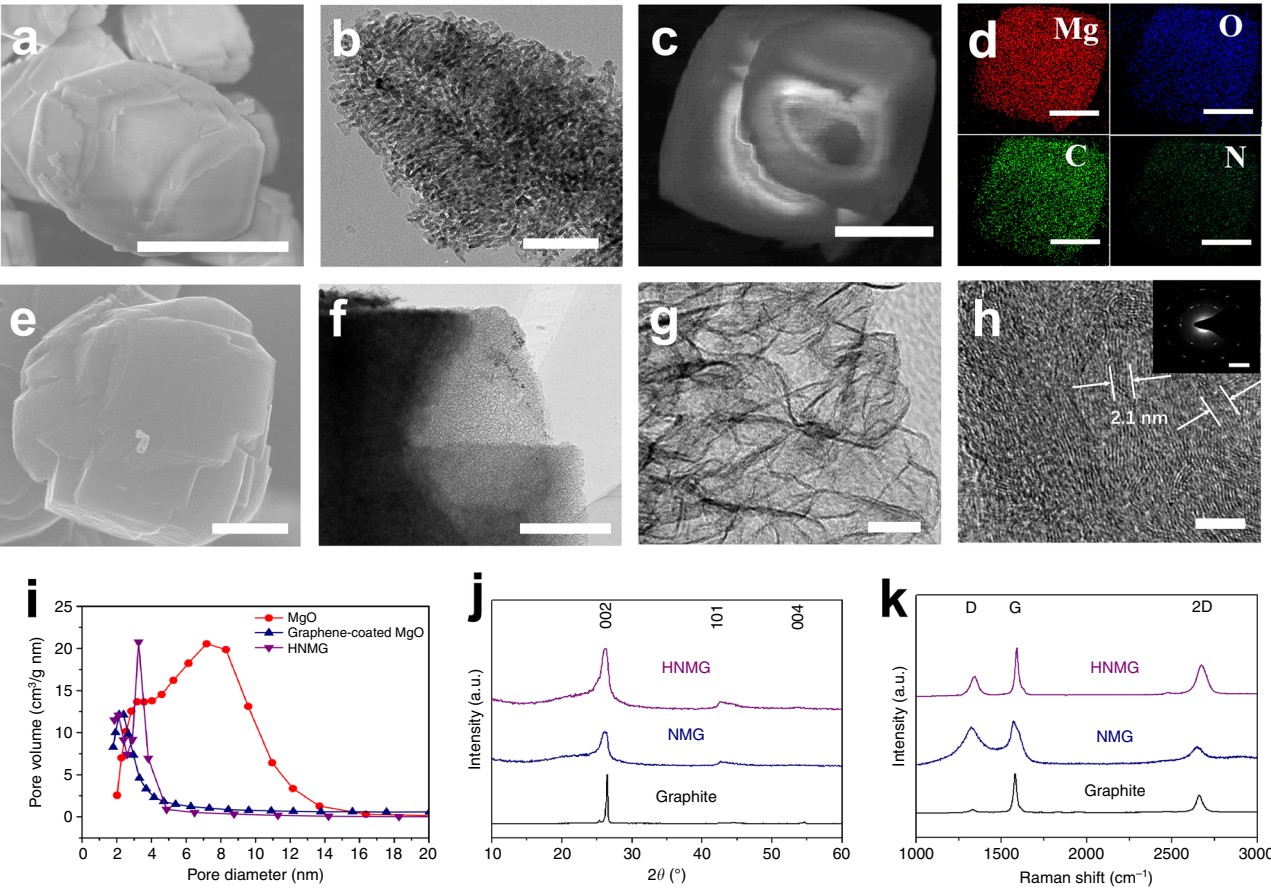

**Fig. 2** Structure of the high-quality nitrogen-doped mesoporous graphene (HNMG) particles. **a, b** SEM and TEM images of mesoporous MgO particles. Scale bars: a 5 μm; **b** 100 nm. **c, d** element mapping of Mg, O, C, and N of a MgO particle after CVD deposition. Scale bars: **c** 2.5 μm; **d** 2.5 μm. **e** SEM, **f, g** TEM, and **h** high-resolution TEM images of the HNMG particles. The inset in **h** is the selected area electronic diffraction (SAED) pattern of the HNMG particle. Scale bars: **e** 2 μm; **f** 1 μm; **g** 20 nm; **h** 5 nm; inset of **h**, 5 1/nm. **i** Pore size distributions of MgO particles, MgO particles with deposited graphene, and HNMG particles. **j** XRD patterns and **k** Raman spectra of HNMG particles, NMG particles, and Graphite

away the template results in the formation of NMG particles with a similar morphology (Supplementary Figure 3). As-made NMG particles were further treated by microwave radiation to reduce the defect density, which was conducted by placing the NMG particles in a vial with argon and subjecting them to a microwave radiation (1000 W for 5 min). As-treated NMG, denoted as HNMG, show a similar particular morphology (see the SEM image in Fig. 2e) with 3D mesoporous channels (see TEM image in Fig. 2f, g), of which the thickness of the pore scaffolds is ~ 2.1 nm (see the higher-magnification TEM image in Fig. 2h). The inset in Fig. 2h shows selected area electron diffraction (SAED) of the sample indicating a crystalline structure. Since the thickness of nitrogen-doped graphene is ~ 0.34 nm, this observation indicates the scaffolds of such mesoporous HNMG particles are consisted with ~ 7 layers of graphene.

Figure 2i shows the pore size distribution of the template particles before and after the CVD process. The template particles show an average pore diameter of 8.0 nm, which is decreased to 2.5 nm after the graphene deposition (Fig. 2i). After the microwave treatment, HNMG particles exhibit an average pore diameter of 3.5 nm and a surface area of 768 m$^2$ g$^{-1}$ (Supplementary Figure 4). In term of the composition, X-ray photon spectra (XPS) of the HNMG particles show a peak of C1s, which can be deconvolved into two components centered at 284.3 and 285.4 eV, representing the sp$^2$ C–sp$^2$ C and N–sp$^2$ C bonds, respectively (Supplementary Figure 5a). A peak for N1s is also observed, which can be deconvolved into two components

centered at 398.2 and 401.1 eV, representing the pyridinic and pyrrolic type of nitrogen, respectively (Supplementary Figure 5b). The amount of nitrogen content in the HNMG particles is estimated as 2.1 atom-%. All these evidences collectively confirm the successful synthesis of nitrogen-doped mesoporous graphene particles.

The role of microwave radiation in restructuring the graphene structure and reducing the defect density was confirmed by X-ray diffraction (XRD) and Raman spectrum. As seen in Fig. 2j, compared with NMG particles, HNMG particles show diffraction peaks with higher intensity and narrower width, which is consistent with the SAED pattern (Fig. 2h and Supplementary Figure 3f). Raman spectra of the HNMG and NMG particles show a D band at ~ 1331 cm$^{-1}$ attributing to the disorder structure and a G band at ~ 1591 cm$^{-1}$ attributing to the highly symmetrical sp$^2$ carbon structure[31,32]. Consistently, the intensity ratio of the D/G bands for the NMG particles (0.98) is decreased to 0.18 for the HNMG particles, confirming microwave radiation does effectively restructure the graphene structure and reduce their defect density. Indeed, the ratio of the D/G band intensity in the HNMG particles is close to that of graphite (~ 0.1)[22], indicating a high-quality graphene structure. The ability to reconstruct the NMG structure can be attributed to their C–N bonds that can effectively absorb microwave radiation[33–36].

**Electrochemical performance.** To examine the lithium-ion storage performance, we first measured the charging and discharging

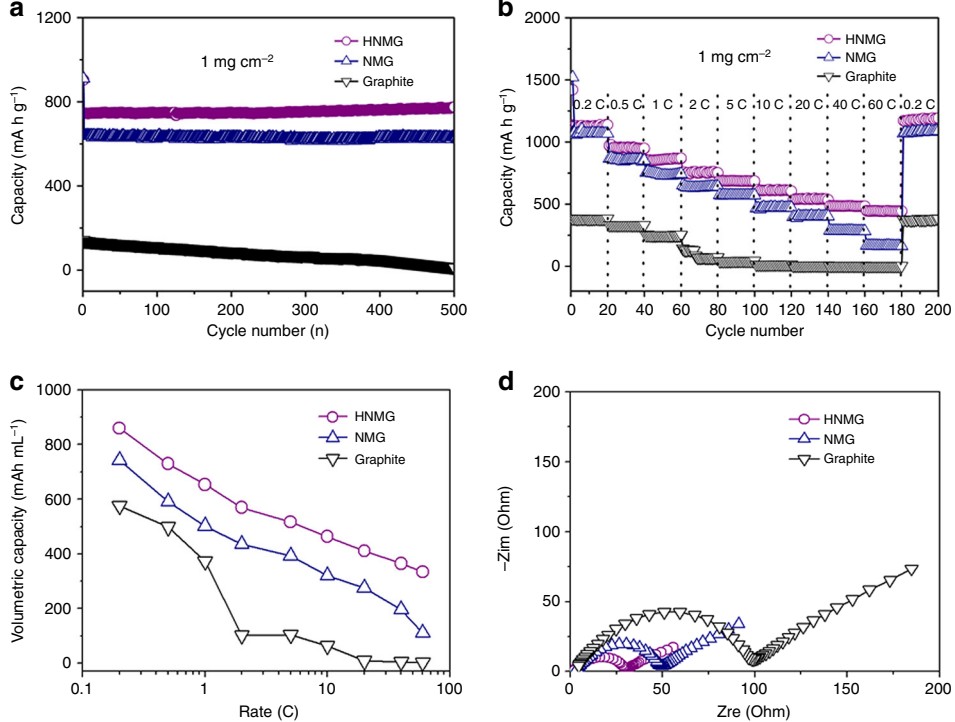

**Fig. 3** Electrochemical performance of the HNMG particles, NMG particles and graphite electrodes. **a** Cycling performance of the HNMG (1 C = 744 mA g$^{-1}$), NMG (1 C = 744 mA g$^{-1}$), and graphite (1 C = 372 mA g$^{-1}$) electrodes with a mass loading of 1 mg cm$^{-2}$ at a rate of 2 C for 500 cycles in the 0.01–2.0 V window (vs. Li/Li$^+$). **b** Rate performance of the HNMG, NMG, and graphite electrodes at various C rates (0.2–60 C). **c** Volumetric capacity of the HNMG, NMG, and graphite electrodes at different C rates. **d** Nyquist plots of the HNMG, NMG, and graphite electrodes prior to the cycling process

capacity and coulombic efficiency. The electrode prepared from HNMG particles with a mass loading of 1 mg cm$^{-2}$ exhibits an initial discharging (lithiation) capacity of 945 mA h g$^{-1}$, charging capacity of 723 mA h g$^{-1}$, and coulombic efficiency of 76.5% at a rate of 2 C (Fig. 3a). The electrode prepared from NMG particles with a same mass loading shows a similar initial discharging capacity of 952 mA h g$^{-1}$, but with a lower initial charging capacity of 612 mA h g$^{-1}$ and lower coulombic efficiency of 64.2%. Generally, the irreversible capacity loss can be attributed to the decomposition of electrolyte, as well as the formation of solid-electrolyte interphase (SEI) on the electrode surface[37,38]. The higher initial coulombic efficiency (ICE) observed in the HNMG electrode can be attributed to the better quality of the graphene scaffolds. After 500 cycles, the HNMG electrode still provides a capacity of 774 mA h g$^{-1}$ at 2 C, which is significantly higher than that of the NMG electrode (628 mA h g$^{-1}$). For comparison, a conventional graphite electrode with a mass loading of 1 mg cm$^{-2}$ was also studied (see Supplementary Figure 6), which shows a much lower specific capacity of 135 mA h g$^{-1}$ in the 2nd cycle and 12 mA h g$^{-1}$ in the 500th cycle at 2 C.

It is worth noting that ICE is a critical parameter for LIBs. Cathode materials (e.g., NMC materials) in LIBs account for a major cost[39], whereas a low ICE could deplete lithium ions, reducing the cell energy density and increasing cost. Compared with commercial graphite[40,41], the HNMG electrode shows a decent ICE of 76.5%, which is significantly higher than the anode materials reported (e.g., N- and S-co-doped graphene with an ICE of 44.7%; graphene aerogels with an ICE of 48.5%; popcorn-like graphene with an ICE of 45.2%; silicon-coated hollow SnO$_2$ nanospheres with an ICE of 62.6%)[9,42–47].

Figure 3b compares the rate performance of the HNMG, NMG, and graphite electrodes at various charge–discharge rates. Increasing the charge–discharge rate from 0.2, 0.5, 1.0, 2, 5, 10,

20 to 40 C, the HNMG electrode exhibits a reversible specific capacity of 1138, 956, 861, 760, 689, 612, 542, and 485 mA h g$^{-1}$, whereas the NMG anode shows a significant lower capacity of 1070, 855, 736, 633, 571, 472, 398, and 282 mA h g$^{-1}$, respectively. Particularly, at a high charge–discharge rate of 60 C, the HNMG electrode still provides a reversible capacity of 448 mA h g$^{-1}$, which is nearly threefold higher than that of the NMG electrode (163 mA h g$^{-1}$) and 70 folds higher than that of the graphite electrode (6 mA h g$^{-1}$). Meanwhile, returning the rate from 60 to 0.2 C, the HNMG electrode shows a high capacity of 1157 mA h g$^{-1}$, indicating an excellent reversibility.

Figure 3c compares the volumetric capacity of the HNMG, NMG, and graphite electrodes with a mass loading of 1 mg cm$^{-2}$. The HNMG anode shows a reversible volumetric capacity of 570 mA h mL$^{-1}$ at 2 C, which is significantly higher than that of the NMG anode (435 mA h mL$^{-1}$) and fivefold higher than that of the graphite electrode (102 mA h mL$^{-1}$). At a higher rate of 60 C, the HNMG anode still shows a volumetric capacity of 334 mA h mL$^{-1}$, which is nearly 3 and 30 folds higher than that of the NMG electrode (110 mA h mL$^{-1}$) and graphite electrode (10 mA h mL$^{-1}$), respectively. The high volumetric capacity observed in the HNMG electrodes can be attributed to the high tap density of the HNMG particles, as well as their high specific capacity and rate performance. Note that such HNMG particles exhibit an average tap density of ∼ 0.63 g cm$^{-3}$, which is one order of magnitude higher than most of the graphene materials reported ( < 0.05 g cm$^{-3}$)[48–50], and only slightly lower than that of the graphite particles used in the conventional anodes (∼ 1.0 g cm$^{-3}$)[51] (Supplementary Table 1). The ability to synthesize such graphene particles with high mass- and volume-specific capacity, as well as high-rate performance, represents an important step toward LIBs with high-energy-power performance.

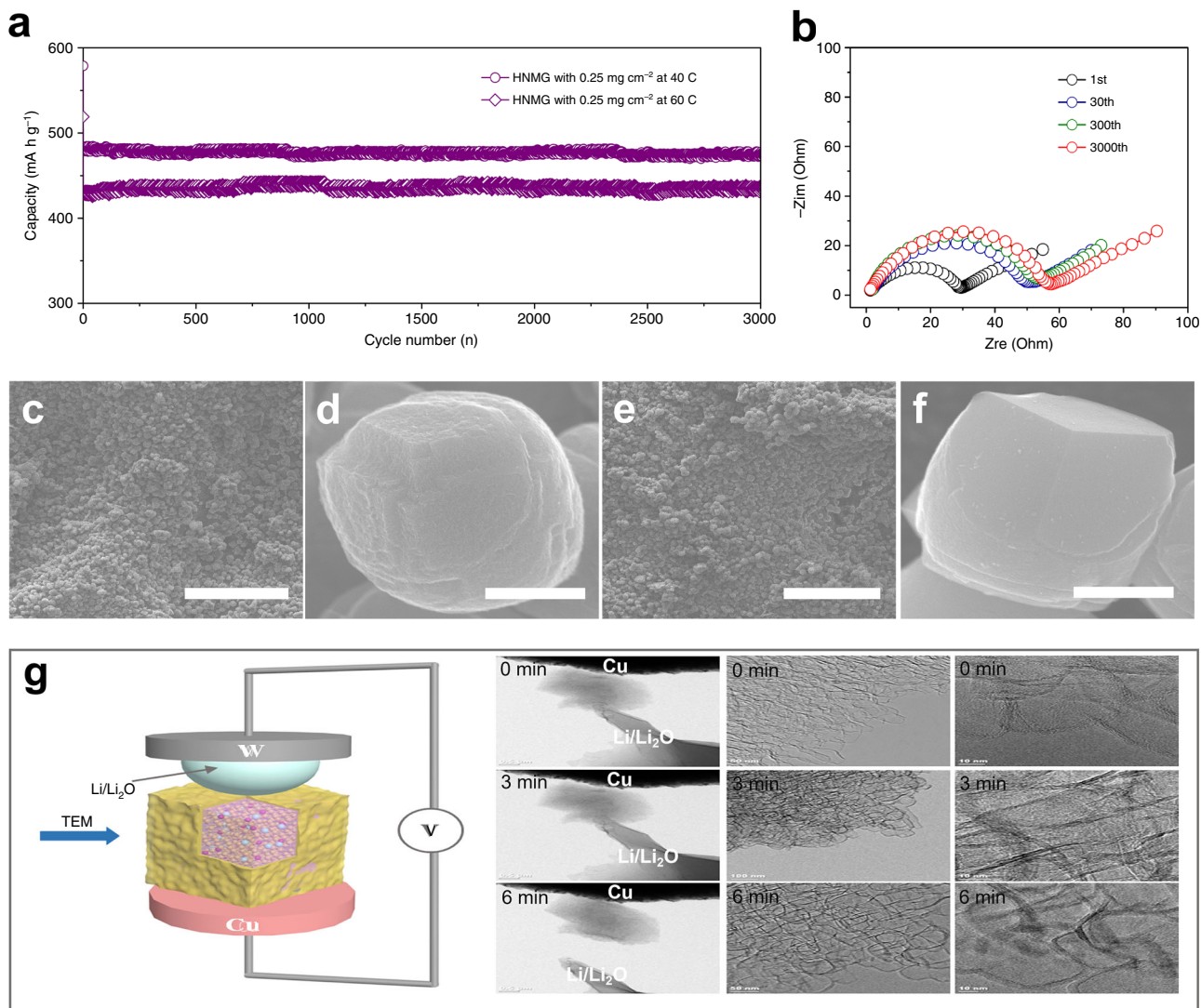

**Fig. 4** The cycling stability of the HNMG electrodes and in situ TEM study of the HNMG particles during charging and discharging. **a** Cycling stability of the HNMG electrode with a mass loading of 0.25 mg cm$^{-2}$ at rates of 40 and 60 C for 3000 cycles in the 0.01–2.0 V window (vs. Li/Li$^+$). **b** Nyquist plots of the HNMG electrode after the 1st, 30th, 300th, and 3000th cycles at a charging and discharging rate of 60 C. **c**, **d** SEM images of a HNMG electrode and HNMG particle prior to the cycling. Scale bars: **c** 100 μm; **d** 3 μm. **e**, **f** SEM images of a HNMG electrode and a HNMG particle in the lithium-insertion state after 3000 cycles at a rate of 60 C. Scale bars: **e** 100 μm; **f** 3 μm. **g** A scheme of an electrochemical cell used for in situ TEM study, as well as the structure and morphology evolution of a HNMG particle during a lithiation and delithiation process

The outstanding rate performance of the HNMG electrodes can be attributed to their effective charge transport, as confirmed by the electrochemical impedance spectroscopic (EIS) study shown in Fig. 3d. Compared with the NMG electrode and the graphite electrode, the HNMG electrode exhibits a shortest Warburg region with the smallest semicircle diameter, indicating fast lithium-ion diffusion with low charge-transfer resistance. It is worth noting that graphene materials with various architectures have been explored for anode applications. Supplementary Figure 7 shows the capacity performance of representative graphene materials reported in the forms of thin films, aerogels, hydrogels, sponges, and foams[52–54]. Compared with these materials, the HNMG particles exhibit a much higher capacity at a given current density. To the best of our knowledge, the high specific capacity of 440 mA h g$^{-1}$ achieved at a high current density of 45 A g$^{-1}$ represents one the best among the graphene materials reported (Supplementary Table 2).

The outstanding rate performance of the HNMG electrodes can be attributed from their fast ion–electron transport.

Supplementary Figure 8 compares the Nyquist plots of a HNMG electrode and a traditional graphite electrode, where the HNMG electrode shows a significantly lower charge-transfer resistance. Note that, during lithiation/delithiation of an electrode, the exchange current density is generally inversely proportional to the charge-transfer resistance[55]. The significantly lower charge-transfer resistance observed for the HNMG electrode indicates a more effective lithiation/delithiation process. Furthermore, the diffusion coefficient of lithium ions within the electrodes was estimated using the Warburg impedance model. As expected, the HNMG electrode exhibits a significantly higher lithium-ion-diffusion coefficient, which is two to three orders of magnitude higher than that of the graphite electrode, further confirming the roles of porous structure in facilitating the ion transport in the electrode (see the Supplementary Information for details, Supplementary Figures 9–12).

Besides their high-power-energy performance, the HNMG electrodes also exhibit outstanding cycling stability. For example, after cycling at 40 and 60 C for 3000 cycles, HNMG electrodes still

show a capacity of 475 and 436 mA h g$^{-1}$ with a capacity retention of 99.2 and 99.1%, respectively (Fig. 4a). Meanwhile, HNMG electrodes exhibit the high coulombic efficiency (Supplementary Figure 13). The NMG electrodes, in contrast, show a significantly lower capacity of 256 and 148 mA h g$^{-1}$ after 500 cycles at 40 and 60 C, respectively (Supplementary Figure 14), whereas the graphite electrode shows a capacity of 7 and 3 mA h g$^{-1}$ after 100 cycles at 40 and 60 C, respectively (Supplementary Figure 15). It is noted that this high areal capacity with high current density under long cycling performance testing will induce dramatically dendrite growth phenomenon on lithium-metal electrodes. Therefore, in order to alleviate the dendrite growth phenomenon on lithium-metal electrodes, HNMG and NMG electrodes with a mass loading of 0.25 mg cm$^{-2}$ were used for the high-rate cycling performance testing, where 4 M lithium bis(fluorosulfonyl)imide in 1,2-dimethoxyethane was used as the electrolyte. The outstanding cycling stability can be attributed to their unique architecture, where the 3D mesoporous structure prevents irreversible stacking of the graphene layers and compensates the volume change that may occur during the charging and discharging, while the low defect density minimizes irreversible electrode reactions. Consistently, EIS of a HNMG electrode after 30 cycles, 300 and 3000 cycles at 60 C shows a similar transport resistance, indicating the conduction networks for ions and electrons are well retained during the cycling process, thus ensuring the cycling stability (Fig. 4b).

To further probe the cycling stability, structure evolution of the HNMG particles during charging and discharging was examined using SEM and in situ TEM. Figure 4c, e shows SEM images of a HNMG electrode before and after 3000 cycles at 60 C, respectively. As shown, the integrity of the electrode is well retained, where the HNMG particles in the electrode remain homogeneously dispersed and connected. This observation suggests the electron-conduction networks in the electrode is well retained after the cycling process, confirming the HNMG architecture can effectively avoid any substantial volume change during charging and discharging. Figure 4d, f further shows the SEM images of a HNMG particle before and after the cycling process, which shows a similar morphology and indicate an excellent structural stability of the particles. It is noteworthy that N-doped graphene is relatively lithiophilic, offering uniform nucleation sites with small nucleation overpotential[56]. Consistently, it was found that the HNMG electrode appears smoother after the cycling process possibly due to the formation of uniform deposition on the particles.

The in situ TEM investigation by applying a voltage bias of 3 V was conducted using a setup illustrated in Fig. 4g. A HNMG particle dispersed on a Cu half-grid was connected to a Li metal electrode that was deposited on a tungsten (W) wire; while the native Li$_2$O layer on the Li electrode was used as the electrolyte. Applying a negative voltage to the Cu grid initiated the lithiation process of the HNMG particle, while the delithiation was started by applying a reversed voltage bias. Figure 4g also shows the structure and morphology evolution of the HNMG particle during lithiation and delithiation (also see the Supplementary Movie 1). Before lithiation (0 min), the HNMG particle exhibits a porous structure similar to that observed in Fig. 2. The morphology, structure, and dimension of the particle remain unchanged after lithiation (3 min) and delithiation (6 min), confirming the structural stability of the particle during lithiation and delithiation. Particularly, the unchanged dimension of the HNMG particles during the charging and discharging preserves the electron- and ion-conduction networks of the electrode, which is essential to ensure the cycling stability.

It is noteworthy that Zhu et al.[57] demonstrated the fabrication of porous carbons for LIB anodes via carbonizing cotton

infiltrated with a MgO precursor. Compared this low-cost and scalable method, the CVD approach could provide HNMG particles with less defect and amorphous carbon moieties, which could significantly improve the electrochemical performance. Various carbon materials have also been synthesized as anodes for LIBs using hard-template methods (Supplementary Table 3). For carbon materials, a high content of sp$^2$ (less sp$^3$) moiety could improve the electronic conductivity and lead to more effective lithium-ion insertion and desertion. The high-quantity HNMG particles enable effective charge transport and provide outstanding mechanical and electrochemical robustness, endowing the electrodes with high areal mass loadings, high reversible capacity, superior rate capability, and remarkable cycling stability. In this context, developing techniques (e.g., microwave irradiation) that can reduce the defect density could be of great interest toward high-performance carbons for LIBs.

Such HNMG particles are ideal materials for anodes with high-energy-power performance and long cycling life. Toward commercial use, it requires the electrodes be fabricated with sufficient areal capacity and current density. This generally requires the electrodes be fabricated with sufficient loading of active materials. However, increasing the mass loading generally increases the transport resistance, which deteriorates the capacity and power performance. Fortunately, the unique architecture of the HNMG particles provide effective transport pathways for ions and electrons, enabling the fabrication of high-mass-loading electrodes with high-energy-power performance.

For demonstration, Fig. 5a shows the charging and discharging curves and specific capacity of the HNMG electrodes with a mass loading of 1, 3, and 6 mg cm$^{-2}$ at the rate of 2 C, respectively. The capacity of these electrodes is mainly contributed by Li insertion at voltage below 0.4 V (vs. Li$^+$/Li), which ensures a high full-cell voltage with high-energy density[22,58]. The electrodes show a capacity of 746, 701, and 642 mA h g$^{-1}$, respectively, at the rate of 2 C (Fig. 5b), indicating the capacity is well retained with increasing the mass loading. Figure 5c further shows the mass-specific capacities of the electrodes at different C rates. With increasing the mass loading from 1 to 3 to 6 mg cm$^{-2}$, the capacity decreases from 1138 to 1126 to 1109 mA h g$^{-1}$ at 0.2 C or from 605 to 535 to 455 mA h g$^{-1}$ at 10 C, respectively, corresponding to a capacity reduction of 1.1 and 2.5% at 0.2 C or 11.6 and 24.8% at 10 C. Nevertheless, even at a high charge–discharge rate of 40 C, the HNMG anodes with a mass loading of 3 and 6 mg cm$^{-2}$ can still provide a capacity of 362 and 221 mA h g$^{-1}$, respectively, confirming the feasibility of fabricating high-mass-loading electrodes with high-energy-power performance. In addition, HNMG anodes with high areal mass loading also exhibit excellent cycling stability. For example, HNMG electrodes with mass loading of 3 mg cm$^{-2}$ can provide a retention of 93% from the initial capacity and a coulombic efficiency of 99.9% (from the second cycle) after 500 cycles at 5 C (Supplementary Figures 16 and 17).

Figure 5d further plots the areal capacity of the HNMG electrodes vs. mass loading at different C rates. The areal capacity linearly increases with mass loading when the C rate is <10 C; further increasing the C rate (e.g., 40 C) deviates the linear relationship at high mass loading (e.g., 6 mg cm$^{-2}$) due to the increased charge resistance and the less effective utilization of the active material. Figure 5e further plots the utilization of the active material vs. the C rate, of which the utilization is estimated by normalizing the specific capacity of the electrode at different C rates (the slopes of the lines) vs. the specific capacity at 0.2 C. The utilization decreases with increasing mass loading and C rate, which is 53, 49, and 44% at 10 C or 48, 42, and 37% at 20 C, for the electrodes with a mass loading of 1, 3, and 6 mg cm$^{-2}$, respectively. It was found that increasing mass loading from 1 to

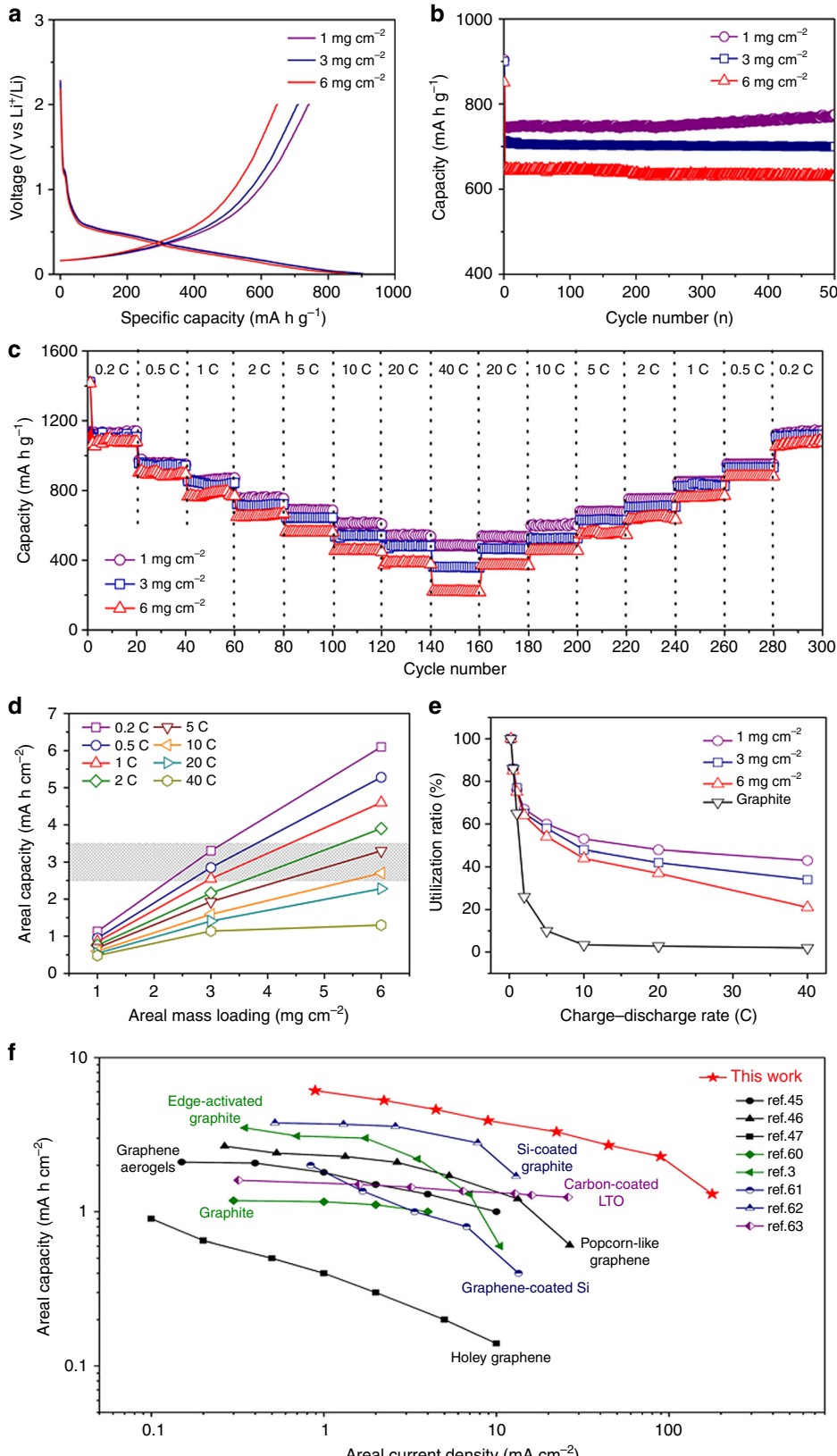

**Fig. 5** The performance of the HNMG electrodes with various mass loadings. **a** Constant current first cycle charge–discharge curves at a rate of 2 C for the HNMG electrodes with a mass loading of 1, 3, and 6 mg cm$^{-2}$, respectively. **b** The capacity of the HNMG electrodes with different mass loadings cycled at a rate of 2 C for 500 cycles in the 0.01–2.0 V window (vs. Li/Li$^{+}$). **c** Specific capacity of the HNMG electrodes with different mass loadings measured at charge–discharge rate from 0.2 to 40 C. **d** Areal capacity and **e** utilization of the active material of the HNMG electrodes with various mass loading at different C rates. The gray area marked in **d** represents the range of the areal capacity of common commercial anodes. **f** A comparison of the areal performance metrics of a HNMG electrode with a mass loading of 6 mg cm$^{-2}$ with representative anodes reported

$6\,mg\,cm^{-2}$ only reduces the utilization for ~10% even at high charge–discharge rates. In contrast, the utilization of the graphite electrode (mass loading of $1\,mg\,cm^{-2}$) rapidly decreases with increasing C rate (3.4% at 10 C or 2.8% at 20 C). These studies further confirm the high-energy-power performance of the high-mass-loading HNMG electrodes.

To further assess the performance of the HNMG electrodes, Fig. 5d also marks the areal capacities of commercial graphite anodes, which are in the range of $2.5$–$3.5\,mA\,h\,cm^{-2}$ when operated at a current-density range of $0.37$–$1.86\,mA\,cm^{-2}$ [59]. As shown, the HNMG electrode with a mass loading of $6\,mg\,cm^{-2}$ exhibits areal capacity as high as 6.1, 5.3, 4.6, 3.9, and $3.3\,mA\,h\,cm^{-2}$ at a charge–discharge rate of 0.2, 0.5, 1, 2, and 5 C, respectively. Such charge–discharge rates correspond to areal current density of 0.9, 2.2, 4.5, 9.0, and $22.3\,mA\,cm^{-2}$, respectively. Compared with the commercial graphite anodes, the areal capacities of the HNMG electrodes are significantly higher, particularly, at high current density.

It is worth noting that various materials have been explored for high-energy-power anodes represented by carbon-coated LTO, silicon-coated graphite, graphene-coated silicon, and structure-engineered graphite and graphene (edge-activated graphite, popcorn-like graphene, graphene aerogel, and holey graphene)[3,45–47,60–63]. Fig. 5f plots the areal capacity vs. areal current density of these materials, along with that of the HNMG electrode and graphite electrode. For the structure-engineered graphene, despite creating holes in the holey graphene facilitate ion transport, their two-dimensional architecture limits the performance at high current density; while graphene aerogel and popcorn-like graphene to possess 3D porous structure, exhibiting higher areal capacity than the graphite anode[45–47]. Activating the edge planes of graphite by a catalytic reaction with hydrogen results in edge-activated graphite, which exhibits better areal capacity than the graphite anode. However, the rate performance at high current density is still intrinsically limited by the lithium insertion/desertion kinetics. For the silicon-containing graphene and graphite, incorporation of high-capacity silicon increases the areal capacity at low current density. Inherently limited by the slow reaction of silicon, however, the capacity is reduced rapidly with increasing current density. In contrast, the HNMG electrodes outperform these representative anodes, exhibiting the highest areal capacity, particularly, at high areal current density. For example, the HNMG anode can supply a high areal capacity over $3\,mA\,h\,cm^{-2}$ with current density of $30\,mA\,cm^{-2}$, which is the best-known values for the anodes reported.

## Discussion

In summary, high-quality graphene electrode for fast-charging and high-energy LIBs have been realized by designing three-dimensional nitrogen-doped mesoporous graphene particles with low density of defects at the nanoscale, which allows us to maximize the potential of graphene for application in fast-charging and high-energy LIBs. The unique architecture of the HNMG particles provides structural and electrochemical stability, effective ion and electron transport, and sufficient sites for ion storage, leading to high-energy capacity and coulombic efficiency, excellent rate performance and long cycling lifetime. Particularly, such HNMG particles enable the fabrication of electrodes with a high areal capacity at high areal current density, providing a highly promising anode material toward LIBs with high-energy-power performance. Importantly, this work opens a pathway to develop the high-quality graphene microstructure with real-life applications in high performance electrochemical energy storage and conversion and beyond.

## Methods

**Preparation of the high-quality nitrogen-doped mesoporous graphene (HNMG) particles.** Overall, 5.4 g of $Mg(NO_3)_2\,6H_2O$ dissolved in 40 ml of water and 2.2 g of $Na_2CO_3$ in 20 ml of water were well mixed and transferred to a Teflon-lined autoclave, which was reacted at 180 °C for 24 h to produce $MgCO_3$. The as-formed $MgCO_3$ was collected by filtration and calcined in air at 600 °C for 2 h to form the mesoporous MgO template particles. The MgO particles in a quartz boat were first placed in the center of a tube furnace with a gas flow containing Ar (500.0 mL min$^{-1}$)/H$_2$ (150.0 mL min$^{-1}$) and heated to 900 °C. Then, another Ar stream (80.0 mL min$^{-1}$) flowing through a flask of acetonitrile at room temperature was introduced to the reactor to grow N-doped graphene within the MgO particles. After deposition for 15.0 min, as-formed particles were collected and treated with hydrochloric acid (1 M) to remove the MgO template and form the NMG particles. The as-prepared NMG particles (500 mg) were then placed in a vial filled with Ar and microwaved (Danby microwave oven, 1000 W) for 5 min to form HNMG particles.

**Material characterization.** Powder X-ray diffraction (XRD) was determined using a Rigaku Miniflex II diffractometer with Cu Kα radiation. The morphology, crystalline phase, and composition of the as-synthesized products were performed on field-emission scanning electron microscopy (FESEM, FEI Nova 430), transmission electron microscopy, and high-resolution transmission electron microscopy (HRTEM, FEI Titan STEM). XPS analysis was characterized using an ESCALAB 250Xi spectrometer by a mono Al Kα radiation. Raman spectroscopy was measured with a Renishaw 2000 System. The pore size distribution and specific surface area were measured using an ASAP 2020 analyzer (Micromeritics Instrument Corporation, Norcross, GA). In situ TEM was carried out using a FEI Titan microscope operated at 300 kV.

**Electrochemical measurements.** To prepared the electrodes, active materials and binder (PVDF) were mixed with a weight ratio of 90:10, which were coated onto a Cu foil. The graphite anode was fabricated by mixing the graphite powder, binder (PVDF), and Super P carbon black with a weight ratio of 80:10:10. These electrodes were integrated into CR2025-type coin cells using Li metal foil as the counter electrode, Celgard 2250 as the separator, and 1 M LiPF$_6$ dissolved in a mixture of dimethyl carbonate (DMC), ethylene carbonate (EC), and diethyl carbonate (DEC) (1:1:1, by volume) as the electrolyte. HNMG electrodes with a mass loading of 0.25 mg cm$^{-2}$ were used for high-rate cycling performance testing, where 4 M lithium bis(fluorosulfonyl)imide in 1,2-dimethoxyethane was used as the electrolyte.

The charge–discharge curves were measured using a Land battery test system (LAND CT2001A) at room temperature. Electrochemical impedance measurements were conducted between 100 kHz and 10 mHz using a perturbation amplitude of 5 mV on the cells at open-circuit potential. In this work, the charge–discharge rates (C rate) are based on HNMG anode (1 C = 744 mA g$^{-1}$), NMG anode (1 C = 744 mA g$^{-1}$), and graphite anode (1 C = 372 mA g$^{-1}$).

## Data availability

The data supporting the findings of this work are available within the article and its Supplementary Information files. All other relevant data supporting the findings of this study are available from the corresponding author on request.

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

## Acknowledgements

We are grateful to the funding support from the UCLA ENN Center for Nanomedicine and Energy Conversion.

## Author contributions

J.L. and G.W. conceived the idea. R.M., F.L., X.T., and R.T. carried out material synthesis, characterization and electrochemical characterization. P.X. constructed the equivalent circuit model. X.W. and C.W. performed the in situ TEM characterization. G.S., X.L., F.L., L.S., B.X., and Q.X. also contributed to the related experiments. R.M. and Y.L. co-wrote the paper. All the authors discussed the results and commented on the manuscript.

## Additional information

**Competing interests:** The authors declare no competing interests.

