## [Peer Review File · Nature Communications]

Reviewers' comments:

Reviewer #1 (Remarks to the Author):

This manuscript reported nitrogen-doped mesoporous graphene particles with extraordinary rate performance, and the author provided many proofs about their rate performances. Since nature communication is the high impact journal, despite rate performances are excellent, but the authors did not provide enough for why and how they work. We feel this work cannot reach the acquirement of nature communication. The specific comments are as following:

1. Likewise, the author said "The mesoporous structure provides lithium-storage sites, effective ion-transport pathways, as well as space to accommodate the volume change", it seems that proofs are not enough and convincing just from simple rate and cycling performance. There is no information for reaction or diffusion kinetics analysis on their materials.
2. The materials was paired with Li foil for high rate cycling performance, for example, 40 C (calculated for 29.76 mA cm⁻²@1 mg cm⁻²) with capacity of ~500 mAh g⁻¹ (calculated for ~0.5 mA cm⁻²) are cycled for 3000 cycles. We are wondering that why conventional Li foil can sustain the high current density of 30.0 mA cm⁻² with the areal capacity of 0.5 mA cm⁻² for stable cycling over 3000 times in unmodified carbonate-based electrolyte? This ultrahigh current density under this areal capacity will induce dramatical dendrite growth phenomenon.
3. Besides the BET measurements for mesoporous morphology, there are no other proofs about how the mesopores work to affect ion-transportation for high-rate ability?
4. As for volume change, although authors provided in-situ TEM for showing the unchanged morphology and volume, we are wondering that during TEM tests, the applied current density maybe too low to show their actual images when they are tested at ultrahigh rate.
5. SEM images showed the cycled particle, we are wonder why the surface of cycled one is smooth than the original one. More importantly, what happened when such high current density and areal capacity applied on nitrogen-doped mesoporous graphene particles?
6. The title about the nitrogen-doped graphene particles, however, we nearly found no information about how doped nitrogen atoms functions and contributes on high current density situations.

Reviewer #2 (Remarks to the Author):

Comments for NCOMMS-18-26600:

Overall, this is an excellent piece of work. The authors demonstrated a simple template method to prepare high-quality graphene electrodes for fast-charging and high-energy lithium ion batteries. The high-quality, nitrogen-doped, mesoporous graphene particles own higher capacity and great rate performance. The stability is also pretty good. The authors also carried out extensive characterization such as in situ TEM to find why the performance is so great. Their explanation is sound. This work is recommended for publication in Nature Communications after minor revision. Some detailed comments are provided as follows:

1. Theoretical capacity of N-doped mesoporous graphene should be different from pure graphene, authors set 1C of HNMG as 744 mA g⁻¹, this should be more careful.
2. The authors think the coulombic efficiency of HNMG is stable and great, but it cannot be found in figure 2A, the coulombic efficiency under 40C or 60C should be added.
3. In this work, the authors have shown that N-doped graphene derived from MgO template could improve the stability of electrode, but there are many articles have reported the similar template method to prepare mesoporous graphene or carbon particles. Electrochemical performance of hard

template method derived electrodes should be listed as comparison.

4. The authors mentioned that MgO template can be used as catalyst, but in the article the catalyst performance of MgO is not clear, please give detailed explanation of the MgO as a catalyst during the preparation of HNMG.

Reviewer #3 (Remarks to the Author):

I found this work to be very exciting and believe it could be a game changer in the lithium-ion battery world. The capacity and rate performance of their HNMG is surprisingly high. If this material is successfully scaled up, it could eliminate the need for silicon-based anodes and perhaps even lithium metal anodes. This paper should be accepted with minor revisions/corrections.

My one major concern regards the lithium metal counter electrode used in this work. The authors show cycling data of 500 to 3,000 cycles with their HNMG electrodes versus lithium metal electrodes, often at very high C-rates (40C to 60C in Figure 3A). What is the material loading (mg/cm² or mAh/cm²) of the HNMG electrodes in Figure 3A? My experience with lithium metal is that you can only achieve high cycle life if the capacity loading of the working electrode is very low. If the electrodes were indeed of a commercial loading (near 3 mAh/cm²), then this implies that the lithium metal electrode can also achieve 3,000 cycles at a fast rate, which I find hard to believe. Why do we need HNMG if lithium metal can do the same thing?

Please include the voltage windows used in the cycling data.

Please specify the material loading for each electrode shown in the figures, especially for the cycle life plots.

Please include the first cycle charge and discharge voltage profiles for one of the HNMG electrodes.

What was the cycle number for the voltage profiles in Fig. 4A?

The following are spelling and grammatical corrections that should be made:

Line 33: "density" can be deleted since "areal capacity" implies per area.

Line 44: I think you should replace "and sluggish of" with "and have relatively slow".

Line 46: "owning" should be "due".

Line 48: insert a space after "above".

Line 50: insert "been" after "also".

Line 52: replace "unsatisfied" with "unsatisfactory".

Line 69: "with nitrogen doping" seems redundant since you state "nitrogen-doped" in line 68.

Line 131: replace "conforming" with "confirming".

Line 139: I recommend adding "(lithiation)" after "discharging".

Line 150: These capacities for graphite seem very low - what is the electrode loading?

Line 168: were the charge and discharge cycles at each rate symmetric (i.e., 40C charge and 40C discharge)?

Line 178: were all three electrodes at 1 mg/cm² loading (graphite too)?

Line 209: delete "in" after "particle".

Line 217: replace "shows" with "show".

Line 226: replace "process and" with "process, thus".

Line 234: replace "effective" with "effectively".

Line 241: how much voltage was needed to cycle lithium in and out of the particle with no electrolyte present?

Line 269: insert "to" after "corresponding".

Line 298: replace "for" with "by".

Line 319: insert "to" after "grapheme".

Line 328: replace "outperformance" with "outperform".

Line 352: insert "a" before "tube".

Lines 352-353: are there two Ar flows (500 and 80 mL/min)? Please correct or explain further.

Line 354: delete "acetonitrile" - repeated word.

Line 358: Was any milling needed to break up the particles/agglomerates?

Line 361: replace "performed on" with "determined using".

Line 365: replace "radiatio" with "radiation".

Response to Reviewer #1:

This manuscript reported nitrogen-doped mesoporous graphene particles with extraordinary rate performance, and the author provided many proofs about their rate performances. Since nature communication is the high impact journal, despite rate performances are excellent, but the authors did not provide enough for why and how they work. We feel this work cannot reach the acquirement of nature communication. The specific comments are as following:

Response/corrections: We sincerely appreciate the Reviewer's insightful comments, which greatly help us better organize and present the results. All the major changes are highlighted in yellow in the revised manuscript. As pointed out by Reviewer #2 and Reviewer #3, this work could significantly impact the industry and the explanation is sound; we believe this technology may contribute significantly to the battery industry. Therefore, the authors hope the novelty and significance of the revised manuscript is suitable for publication.

1. Likewise, the author said "The mesoporous structure provides lithium-storage sites, effective ion-transport pathways, as well as space to accommodate the volume change", it seems that proofs are not enough and convincing just from simple rate and cycling performance. There is no information for reaction or diffusion kinetics analysis on their materials.

Response: We thank the reviewer for this comment and agree that reaction or diffusion kinetics analysis on HNMG materials should be added. Regarding the roles of porous structure in mass transport and chemical reaction, it is a common understanding that porous structure does provide surface sites and transport pathways in the science community. In term of using voids (pores) to accommodate the volume change of electrode materials during the charging and discharging, it is also a common strategy used in the battery community. To further address this comment, EIS studies were conducted on electrodes based on our mesoporous graphene and non-porous graphite. As expected, significantly higher ion-diffusion coefficient and lower charge-transfer resistance are observed for the mesoporous graphene electrode, which is consistent with the excellent rate performance observed. We have provided this information in the revised manuscript.

Please see **page 8, line 28- page 9, line 8** in the revised manuscript:

"The outstanding rate performance of the HNMG electrodes can be attributed from their fast ion-electron transport. Fig. S8 compares the Nyquist plots of a HNMG electrode and a traditional graphite electrode, where the HNMG electrode shows a significantly lower charge-transfer resistance. Note that, during lithiation/delithiation of an electrode, the exchange current density is generally inversely proportional to the charge-transfer resistance.⁵³ The significantly lower charge-transfer resistance observed for the HNMG electrode indicates a more effective lithiation/delithiation process. Furthermore, the diffusion coefficient of lithium ions within the electrodes was estimated using the Warburg impedance model.⁵⁴ As expected, the

HNMG electrode exhibits a significantly higher lithium-ion-diffusion coefficient, which is two to three orders of magnitude higher than that of the graphite electrode, further confirming the roles of porous structure in facilitating the ion transport in the electrode (See the Supplementary Information for details, Fig. S9-Fig. S12).”

Supplementary Figure 8. Nyquist plots of a HNMG and a graphite anode obtained by applying a sine wave with amplitude of 5.0 mV over the frequency range from 100 kHz to 0.01 Hz and an equivalent circuit used to fit the Nyquist plot. The circuit elements are composed of the solution resistance (R_s), the charge-transfer resistance (R_{ct}), the contact resistance (R_f) and the Warburg impedance (Z_w), respectively.

Supplementary Figure 9. The relationship between the real part of the impedance spectra (Z_{re}) and $\omega^{-1/2}$ in the low-frequency region, where ω is the angular frequency in the low-frequency region, $\omega = 2\pi f$.

2. The materials was paired with Li foil for high rate cycling performance, for example, 40 C (calculated for 29.76 mA cm^{-2} @ 1 mg cm^{-2}) with capacity of $\sim 500 \text{ mA h g}^{-1}$ (calculated for $\sim 0.5 \text{ mA h cm}^{-2}$) are cycled for 3000 cycles. We are wondering that why conventional Li foil can sustain the high current density of 30.0 mA cm^{-2} with the areal capacity of 0.5 mA h cm^{-2} for stable cycling over 3000 times in unmodified carbonate-based electrolyte? This ultrahigh current density under this areal capacity will induce dramatical dendrite growth phenomenon.

Response: We sincerely appreciate the reviewer's important comments. As reviewer rightly pointed out, this ultrahigh current density under high areal capacity will induce dramatically dendrite growth phenomenon. In our work, the mas loading of the HNMG electrode in this test is 0.25 mg cm^{-2} (rather than 1 mg cm^{-2}). Therefore, the areal capacity of such electrodes is $0.125 \text{ mA h cm}^{-2}$ (rather than 0.5 mA h cm^{-2}).

To demonstrate that such cells can be operated under such a current density, Li||Li cells, as well as Li||HNMG cells, were assembled with separators having a hole at the center to allow effective penetration of dendrites. Li plating/stripping was conducted at a high current density of 45 mA cm^{-2} . With a plating/stripping capacity of $0.125 \text{ mA h cm}^{-2}$ per cycle, the Li||Li cell showed increasing voltage hysteresis and was shorted after 6,000 cycles. For the Li||HNMG cell, the voltage hysteresis was significantly smaller and the voltage remained unchanged for more than 6,000 cycles (**Fig 1**). Furthermore, the Li||HNMG cell could be cycled at a current density of 45 mA cm^{-2} and a higher capacity of $0.25 \text{ mA h cm}^{-2}$ for 3,000 cycles without shorting (**Fig 2**).

The mitigated growth of dendrite of the Li||HNMG cell is associated with the unique structure and composition of the HNMG electrodes. For example, the high surface area could reduce the local current density; while the nitrogen doping makes the HNMG lithiophilic with sufficient nucleation sites and low nucleation overpotential.

Figure 1. Galvanostatic cycling of two different types of symmetric cells which are composed of Li-metal foil (black line), and Li-HNMG scaffold (red line). The current density was fixed at 45 mA cm^{-2} with each cycle set to 10 s (areal capacity of $0.125 \text{ mA h cm}^{-2}$).

Figure 2. Galvanostatic cycling of a symmetric Li-HNMG scaffold cell (red line). The current density was fixed at 45 mA cm^{-2} with each cycle set to 20 s (areal capacity of $0.25 \text{ mA h cm}^{-2}$).

Please see **page 16, line 26- line 28 in the revised manuscript:**

“HNMG electrodes with a mass loading of 0.25 mg cm^{-2} were used for high-rate cycling performance testing, where 4 M lithium bis(fluorosulfonyl) imide in 1,2-dimethoxyethane was used as the electrolyte.”

3. Besides the BET measurements for mesoporous morphology, there are no other proofs about how the mesopores work to affect ion-transportation for high-rate ability?

Response: We thank the reviewer for this valuable suggestion. It is commonly known that, porous structure, particularly, pore size and tortuosity, can dramatically affect mass transport. To address this comment, EIS studies were conducted on a HNMG electrode and a graphite electrode. It was found that the HNMG electrode exhibits a significantly higher lithium-ion-diffusion coefficient, which is two to three orders of magnitude higher than that of the graphite electrode; meanwhile, the charge-transfer resistance of the HNMG electrode is significantly lower than that of the non-porous graphite electrode. This study confirms the mesoporous structure facilitate ion transport with reduced charge-transport resistance. We have provided this information in the revised manuscript.

Please see **page 8, line 28- page 9, line 8** in the revised manuscript:

“The outstanding rate performance of the HNMG electrodes can be attributed from their fast ion-electron transport. Fig. S8 compares the Nyquist plots of a HNMG electrode and a traditional graphite electrode, where the HNMG electrode shows a significantly lower charge-transfer resistance. Note that, during lithiation/delithiation of an electrode, the exchange current density is generally inversely proportional to the charge-transfer resistance.⁵³ The

significantly lower charge-transfer resistance observed for the HNMG electrode indicates a more effective lithiation/delithiation process. Furthermore, the diffusion coefficient of lithium ions within the electrodes was estimated using the Warburg impedance model.⁵⁴ As expected, the HNMG electrode exhibits a significantly higher lithium-ion-diffusion coefficient, which is two to three orders of magnitude higher than that of the graphite electrode, further confirming the roles of porous structure in facilitating the ion transport in the electrode (See the Supplementary Information for details, Fig. S9-Fig. S12).”

Please see **page 7, line 6- page 8, line 8** in the revised supplementary information:

“According to the Warburg impedance model, $Z_{re} = \delta \omega^{-1/2}$, where Z_{re} is the real part of the resistance, δ is the Warburg prefactor and ω is the angular frequency.²⁸ δ is related to the diffusion coefficient of lithium ions (D) by

$$\delta = \frac{V_m \frac{dE_{oc}}{dx}}{FA(2D)^{0.5}}$$

where V_m is the molar volume of the lithiated HNMG or graphite, F is the Faraday constant, A is the electrode area, and dE_{oc}/dx is the gradient of the coulometric titration curve. dE_{oc}/dx can be obtained from a plot of the open-circuit potential (E_{oc}) vs. the molar fraction of lithium “ x ” in the HNMG or graphite at each charged state.

Assuming that graphite and graphene have a similar molar volume (V_m) and that both the electrode have a similar electrode area (A), and F is the Faraday constant, therefore, δ and dE_{oc}/dx related to the diffusion coefficient of lithium ions (D) by

$$\frac{D(HNMG)}{D(Graphite)} = \left[\frac{\frac{dE_{oc}}{dx(HNMG)}}{\frac{dE_{oc}}{dx(Graphite)}} \frac{\delta(Graphite)}{\delta(HNMG)} \right]^2$$

Based on the δ measured ($16 \Omega/s^{0.5}$ and $64 \Omega/s^{0.5}$ for the HNMG and graphite electrode, respectively), the ratio of δ between the HNMG and the graphite electrode is 1/4. Based on the results shown in Fig. S9-Fig. S12, the ratio of dE_{oc}/dx for the HNMG electrode vs the graphite electrode varies in the range of 5 to 15. Accordingly, the ratio of the diffusion coefficient of lithium ion in the HNMG and graphite electrode, $[\frac{D(HNMG)}{D(Graphite)}]$, can be estimated with a number ranging from 400 to 3600. The diffusion coefficient of lithium ion in the HNMG electrode, based on the calculation above, is 2 to 3 orders of magnitude higher than that of the graphite electrode.”

Supplementary Figure 9. The relationship between the real part of the impedance spectra (Z_{re}) and $\omega^{-1/2}$ in the low-frequency region, where ω is the angular frequency in the low-frequency region, $\omega = 2\pi f$.

Supplementary Figure 10. Plots of the open-circuit potential (E_{oc}) of a HNMG and a graphite electrode vs. their molar fraction of lithium (x) in the lithiated electrodes (Li_xC).

Supplementary Figure 11. The gradient of the coulometric titration curve (dE_{oc}/dx) vs. the lithium composition x in the lithiated HNMG and graphite electrodes.

Supplementary Figure 12. A plot of the ratio of $dE_{oc}/dx_{(HNMG)}$ and $dE_{oc}/dx_{(Graphite)}$ vs. the composition x in the lithiated HNMG and graphite electrodes.

4. As for volume change, although authors provided in-situ TEM for showing the unchanged morphology and volume, we are wondering that during TEM tests, the applied current density maybe too low to show their actual images when they are tested at ultrahigh rate.

Response: We sincerely appreciate the reviewer's time and important comments. It should be

noted that the *in situ* lithiation-delithiation via TEM is tested by applying a voltage bias of 3 V. Specifically, the HNMG were loaded on a Cu tip and then connected to Li/Li₂O on a W tip. The native Li₂O on the Li surface served as a solid electrolyte. Lithiation of HNMG started when a negative voltage was applied to the Cu end, while delithiation was initiated upon reversing the sign of the voltage bias. As shown in Fig. 3G, the lithiation time of a HNMG structure is 3 min, corresponding to the high charge/discharge rate of 20 C. And no significant change of the structure and morphology can be observed. Therefore, we think that these experiments allow us to directly observe the mechanical robustness of HNMG electrode during battery operation.

Please see **page 11, line 3- line 15** in the revised manuscript:

“The *in situ* TEM investigation by applying a voltage bias of 3 V was conducted using a setup illustrated in Fig 3G. A HNMG particle dispersed on a Cu half-grid was connected to a Li metal electrode that was deposited on a tungsten (W) wire; while the native Li₂O layer on the Li electrode was used as the electrolyte. Applying a negative voltage to the Cu grid initiated the lithiation process of the HNMG particle, while the delithiation was started by applying a reversed voltage bias. Fig 3G also shows the structure and morphology evolution of the HNMG particle during lithiation and delithiation (also see the Supplementary Movie 1). Before lithiation (0 min), the HNMG particle exhibits a porous structure similar to that observed in Figure 1. The morphology, structure, and dimension of the particle remain unchanged after lithiation (3 min) and delithiation (6 min), confirming the structural stability of the particle during lithiation and delithiation. Particularly, the unchanged dimension of the HNMG particles during the charging and discharging preserves the electron- and ion- conduction networks of the electrode, which is essential to ensure the cycling stability.”

5. SEM images showed the cycled particle, we are wonder why the surface of cycled one is smooth than the original one. More importantly, what happened when such high current density and areal capacity applied on nitrogen-doped mesoporous graphene particles?

Response: We sincerely appreciate the reviewer’s time and important comments. This is an excellent question. It is noteworthy that the surface of N-doped graphene is full of lithiophilic functional groups, offering uniform nucleation sites and small nucleation overpotential (*Angew. Chem. Int. Ed.* 56, 7764-7768, 2017). After long-term cycles, the surface of HNMG electrode in lithium intercalation state can be covered with uniform and smooth Li deposits. Therefore, the surface of cycled one is smoother than the original one. And we have provided this information in the revised manuscript.

Please see **page 10, line 28- page 11, line 2** in the revised manuscript:

“Fig. 3D and 3F further show the SEM images of a HNMG particle before and after the cycling process, which show a similar morphology and indicate an excellent structural stability of the particles. It is noteworthy that N-doped graphene is relatively lithiophilic, offering uniform

nucleation sites with small nucleation overpotential.⁵⁵ Consistently, it was found that the HNMG electrode appears smoother after the cycling process possibly due to the formation of uniform deposition on the particles.”

“55. Zhang, R. et al. Lithiophilic sites in doped graphene guide uniform lithium nucleation for dendrite-free lithium metal anodes. **Angew. Chem. Int. Ed.** 56, 7764-7768 (2017).”

6. The title about the nitrogen-doped graphene particles, however, we nearly found no information about how doped nitrogen atoms functions and contributes on high current density situations.

Response: We thank the Reviewer for highlighting this important point. The roles of nitrogen doping on energy-storage performance of graphene have been well documented. We have provided this information in the revised manuscript.

Please see **page 3, line 15- line 18** in the revised manuscript:

“(4) Nitrogen doping improves electrode-electrolyte interactions, provides lithiophilic surface moieties, and offers uniform nucleation sites with small nucleation overpotential, which has advantages in high energy density with fast-charging capability.”

Response to Reviewer #2:

Overall, this is an excellent piece of work. The authors demonstrated a simple template method to prepare high-quality graphene electrodes for fast-charging and high-energy lithium ion batteries. The high-quality, nitrogen-doped, mesoporous graphene particles own higher capacity and great rate performance. The stability is also pretty good. The authors also carried out extensive characterization such as in situ TEM to find why the performance is so great. Their explanation is sound. This work is recommended for publication in Nature Communications after minor revision. Some detailed comments are provided as follows:

1. Theoretical capacity of N-doped mesoporous graphene should be different from pure graphene, authors set IC of HNMG as 744 mA g⁻¹, this should be more careful.

Response: We sincerely appreciate the reviewer's time and extremely valuable comments. As pointed out, surface modification can enhance the Li storage capability of graphene materials. For example, Wang et al.^[1] reported that nitrogen-doped graphene nanosheets exhibit a high reversible capacity up to 900 mA h g⁻¹. Ma et al.^[2] investigated lithium storage of N-doped graphene using first-principle calculations. The calculation suggests the capacity is depended on the doping structure, which is 1262, 1198 and 1087 mA h g⁻¹ for the pyridinic, pyrrolic and graphitic structures, respectively. In this context, C-rate is commonly used to reflect the electrochemical performance of such materials.^[1,3] For consistency, C-rate is also adapted as a performance index in this work (1C = 744 mA g⁻¹).

[1] Wang, H. et al. Nitrogen-doped graphene nanosheets with excellent lithium storage properties.

J. Mater. Chem. 21, 5430-5434 (2011).

[2] Ma, C. C., Shao, X. H. & Cao, D. P. Nitrogen-doped graphene nanosheets as anode materials for lithium ion batteries: a first-principles study. **J. Mater. Chem.** 22, 8911-8915 (2012).

[3] Wang, X. et al. Atomistic origins of high rate capability and capacity of N-doped graphene for lithium storage. **Nano Lett.** 14, 1164-1171 (2014).

2. The authors think the coulombic efficiency of HNMG is stable and great, but it cannot be found in figure 3A, the coulombic efficiency under 40C or 60C should be added.

Response: We thank the reviewer for this comment and agree that the coulombic efficiency under 40 C or 60 C should be added. We have provided this information in the revised manuscript.

Please see **page 10, line 6- line 7** in the revised manuscript:

“For example, after cycling at 40 C and 60 C for 3000 cycles, HNMG electrodes still show a capacity of 475 and 436 mA h g⁻¹ with a capacity retention of 99.2 % and 99.1%, respectively (Fig. 3A). Meanwhile, HNMG electrodes exhibit the high coulombic efficiency (Fig. S13).”

Supplementary Figure 13. Coulombic efficiency for HNMG electrodes with a mass loading of 0.25 mg cm^{-2} at rates of 40 C and 60 C for 3000 cycles.

3. In this work, the authors have shown that N-doped graphene derived from MgO template could improve the stability of electrode, but there are many articles have reported the similar template method to prepare mesoporous graphene or carbon particles. Electrochemical performance of hard template method derived electrodes should be listed as comparison.

Response: Thank you! We have provided the electrochemical performance of carbon synthesized by hard template methods (Table S3). As shown in the Table, the HNMG does offer outstanding performance compared with other hard-template carbons reported.

Please see **page 10, line 19- line 21** in the revised manuscript:

“Considering various carbon materials have been synthesized using hard-template methods, their electrochemical performance is compared with that of the HNMG particles (Table S3). Clearly, the HNMG particles show outstanding performance.”

Table S3. Electrochemical performance of reported typical various carbon-based materials prepared by template method in comparison with our results.

Materials	Template used	Mass loading (mg cm^{-2})	Current density (A g^{-1})	Capacity (mA h g^{-1})	Current density (A g^{-1})	Capacity (mA h g^{-1})	Reference
Graphene Ball	SiO_2	NA	0.07	700	7	240	[4]
Mesoporous Graphene Sheet	AAO	NA	0.1	770	5	255	[13]
3D Macroporous Carbon Monolith	PMMA	NA	0.015	299	0.15	125	[20]

Hollow Carbon Sphere	SiO ₂	NA	0.074	320	5.58	210	[21]
Hierarchically Porous Carbon Monolith	SiO ₂	2	0.372	540	22.32	70	[22]
Ordered Multimodal Porous Carbon Sphere	SiO ₂	NA	0.1	903	1	758	[23]
Hollow Mesoporous Carbon Sphere	SiO ₂	NA	0.074	268	3.72	100	[24]
Nitrogen-Rich Mesoporous Carbon Plate	CaCO ₃	1	0.1	900	0.8	400	[25]
Nitrogen-Rich Porous Carbon Sphere	SiO ₂	NA	0.5	542	3	215	[26]
Porous Carbon Fiber	MgO	NA	0.25	1020	4	355	[27]
HMNG Particles	0	1	0.15	1138	45	440	This work
		3		1126	30	361	
		6		1109	30	221	

4. The authors mentioned that MgO template can be used as catalyst, but in the article the catalyst performance of MgO is not clear, please give detailed explanation of the MgO as a catalyst during the preparation of HNMG.

Response: We thank the Reviewer for highlighting this important point. The growth of graphene is believed through free-radical condensation (FRC) of hydrocarbons on the MgO surface.²⁹ Compared with graphene from metal catalysts, graphene made using a MgO catalyst generally contains more defects due to insufficient removal of amorphous carbon formed during a FRC process.³⁰ Accordingly, microwave radiation was adapted to decrease the density of defects in this work. We have provided this information in the revised manuscript.

Please see **page 3, line 2- line 6** in the revised manuscript:

“It is believed that the growth of graphene on MgO is achieved through by free-radical condensation of hydrocarbons.²⁹ Compared with metallic catalysts, MgO generally leads to graphene with higher defect density,³⁰ which can be reduced by subsequent microwave radiation enabling the synthesis of high-quality, nitrogen-doped, mesoporous graphene (HNMG) particles.”

“29. Rummeli, M. et al. Direct low-temperature nanographene CVD synthesis over a dielectric insulator. *ACS Nano*. **4**, 4206-4210 (2010).

30. Zhao, J. et al. Synthesis of large-scale undoped and nitrogen-doped amorphous graphene on MgO substrate by chemical vapor deposition. *J. Mater. Chem.* **22**, 19679-19683 (2012).”

Response to Reviewer #3:

I found this work to be very exciting and believe it could be a game changer in the lithium-ion battery world. The capacity and rate performance of their HNMG is surprisingly high. If this material is successfully scaled up, it could eliminate the need for silicon-based anodes and perhaps even lithium metal anodes. This paper should be accepted with minor revisions/corrections.

1. My one major concern regards the lithium metal counter electrode used in this work. The authors show cycling data of 500 to 3,000 cycles with their HNMG electrodes versus lithium metal electrodes, often at very high C-rates (40C to 60C in Figure 3A). What is the material loading (mg/cm^2 or $\text{mA h}/\text{cm}^2$) of the HNMG electrodes in Figure 3A? My experience with lithium metal is that you can only achieve high cycle life if the capacity loading of the working electrode is very low. If the electrodes were indeed of a commercial loading (near $3 \text{ mA h}/\text{cm}^2$), then this implies that the lithium metal electrode can also achieve 3,000 cycles at a fast rate, which I find hard to believe. Why do we need HNMG if lithium metal can do the same thing?

Response: We sincerely appreciate the reviewer's important comments. As pointed out, a high current density could induce dendrite growth. The electrode used in this study has a mass loading of 0.25 mg cm^{-2} , which allowed us to conduct these studies. Please also refer to the Response for the Comment #2, Reviewer 1. The related information has been incorporated to the revised manuscript.

To demonstrate that such cells can be operated under such a current density, Li||Li cells, as well as Li||HNMG cells, were assembled with separators having a hole at the center to allow effective penetration of dendrites. Li plating/stripping was conducted at a high current density of 45 mA cm^{-2} . With a plating/stripping capacity of $0.125 \text{ mA h cm}^{-2}$ per cycle, the Li||Li cell showed increasing voltage hysteresis and was shorted after 6,000 cycles. For the Li||HNMG cell, the voltage hysteresis was significantly smaller and the voltage remained unchanged for more than 6,000 cycles (**Fig 1**). Furthermore, the Li||HNMG cell could be cycled at a current density of 45 mA cm^{-2} and a higher capacity of $0.25 \text{ mA h cm}^{-2}$ for 3,000 cycles without shorting (**Fig 2**).

The mitigated growth of dendrite of the Li||HNMG cell is associated with the unique structure and composition of the HNMG electrodes. For example, the high surface area could reduce the local current density; while the nitrogen doping makes the HNMG lithiophilic with sufficient nucleation sites and low nucleation overpotential.

Figure 1. Galvanostatic cycling of two different types of symmetric cells which are composed of Li-metal foil (black line), and Li-HNMG scaffold (red line). The current density was fixed at 45 mA cm^{-2} with each cycle set to 10 s (areal capacity of $0.125 \text{ mA h cm}^{-2}$).

Figure 2. Galvanostatic cycling of a symmetric Li-HNMG scaffold cell (red line). The current density was fixed at 45 mA cm^{-2} with each cycle set to 20 s (areal capacity of $0.25 \text{ mA h cm}^{-2}$).

Please see **page 16, line 26- line 28 in the revised manuscript:**

“HNMG electrodes with a mass loading of 0.25 mg cm^{-2} were used for high-rate cycling performance testing, where 4 M lithium bis(fluorosulfonyl) imide in 1,2-dimethoxyethane was used as the electrolyte.”

2. Please include the voltage windows used in the cycling data.

Response: We thank the reviewer for this valuable suggestion. In the revised manuscript, we have provided the voltage windows used in the cycling data.

3. Please specify the material loading for each electrode shown in the figures, especially for the cycle life plots.

Response: Thank you! In the revised manuscript, we have added the material loading for each electrode shown in the figures.

4. Please include the first cycle charge and discharge voltage profiles for one of the HNMG electrodes.

Response: Thank you! We have provided the first cycle charge and discharge voltage profiles for one of the HNMG electrodes in the revised manuscript (see Figure 4A).

5. What was the cycle number for the voltage profiles in Fig. 4A?

Response: The cycle number for the voltage profiles in Fig. 4A is the first cycle. This information has been updated in the revised manuscript.

6. The following are spelling and grammatical corrections that should be made:

Line 33: "density" can be deleted since "areal capacity" implies per area.

Line 44: I think you should replace "and sluggish of" with "and have relatively slow".

Line 46: "owning" should be "due".

Line 48: insert a space after "above".

Line 50: insert "been" after "also".

Line 52: replace "unsatisfied" with "unsatisfactory".

Line 69: "with nitrogen doping" seems redundant since you state "nitrogen-doped" in line 68.

Line 131: replace "conforming" with "confirming".

Line 139: I recommend adding "(lithiation)" after "discharging".

Line 150: These capacities for graphite seem very low - what is the electrode loading?

Response: Thank you for your time and efforts for correcting the spelling and grammar! We should and will improve our language capability continuously. For Line 150, the mass loading of the graphite electrode is 1 mg cm^{-2} . This information has been updated in the revised manuscript.

Line 168: were the charge and discharge cycles at each rate symmetric (i.e., 40C charge and 40C discharge)?

Response: The charge and discharge cycles at each rate were symmetric (i.e., 40 C charge and 40 C discharge). Thank you!

Line 178: were all three electrodes at 1 mg/cm^2 loading (graphite too)?

Response: The mass loading of all three electrodes, including the graphite electrode, was 1 mg cm^{-2} . This information has been updated in the revised manuscript. Thank you!

Line 209: delete "in" after "particle".

Line 217: replace "shows" with "show".

Line 226: replace "process and" with "process, thus".

Line 234: replace "effective" with "effectively".

Line 241: how much voltage was needed to cycle lithium in and out of the particle with no electrolyte present?

Response: *In situ* lithiation-delithiation via TEM was tested by applying a constant voltage bias of 3 V. This information has been updated in the revised manuscript.

Line 269: insert "to" after "corresponding".

Line 298: replace "for" with "by".

Line 319: insert "to" after "grapheme".

Line 328: replace "outperformance" with "outperform".

Line 352: insert "a" before "tube".

Lines 352-353: are there two Ar flows (500 and 80 mL/min)? Please correct or explain further.

Response: The MgO particles in a quartz boat were firstly placed in the center of a tube furnace heated to 900 °C with flowing a gas mixture of Ar (500.0 mL min⁻¹)/H₂ (150.0 mL min⁻¹). Then, a stream of Ar (80.0 mL min⁻¹) flowing through a flask of acetonitrile was introduced to the reactor to grow N-doped graphene under 900 °C. This information has been updated in the revised manuscript.

Line 354: delete "acetonitrile" - repeated word.

Line 358: Was any milling needed to break up the particles/agglomerates?

Response: There was no milling process applied. Thank you!

Line 361: replace "performed on" with "determined using".

Line 365: replace "radiatio" with "radiation".

Response: The spelling and grammatical corrections have been made in the revised manuscript. Thank you!

Reviewers' comments:

Reviewer #1 (Remarks to the Author):

The authors do some real and beautiful revisions. But I still think the manuscript haven't addressed the concerns on the lithium metal anodes. Like the reviewer 3 said, why do we need HNMG if lithium metal can do the same thing? In the response letter, the evidences for Li metal anodes seems to be very convincing, but there are actually not conformal with the content in manuscript. The manuscript stressed the outstanding rate capability and high areal capacity of HNMG in the abstract, but as to the comments on lithium metal anodes, especially in the response letter, they said their mass loading was 0.25 mg cm⁻² (In Figure 4B, the areal mass loading was 1-6 mg cm⁻², so the authors did their electrochemical performances tests under quite different conditions?), by the way, I think most readers will consider this ultra-low mass loading is not sufficient for providing high rate capability. And the electrolyte is different for lithium ion batteries and lithium metal anodes, the authors chose the 4M high concentrated ether-based electrolyte, which will have great improvement from normal carbonate-based electrolyte (for Li-ion batteries in manuscript). Although the other two reviewers suggested minor revisions, but I still think this manuscript is not suitable for nature communications.

Reviewer #2 (Remarks to the Author):

Comments for NCOMMS-18-26600A:

The authors had responded most of my concerns properly, some other minor suggestions before accepted:

(1) The authors have added the content about the contribution of MgO as catalyst during the graphene growing, the references such as " [27] Zhu, C. Y. & Akiyama, T. Cotton derived porous carbon via an MgO template method for high performance lithium ion battery anodes. Green Chem. 18, 2106-2114 (2016)" also mentioned the similar method to prepare LIB anode, the authors should show the different between them , for example, crystallinity, CVD method and so on.

(2) The authors make lists to compare electrodes prepared via different template methods, the performance of this work shows the best. The authors said that the outstanding rate performance of the HNMG electrodes can be attributed to their effective charge transport. Thus, the authors should give more detailed explanations or references of the relationship between charge transport and defects, are the better performance just due to the reduction through microwave?

Reviewer #3 (Remarks to the Author):

Thank you for the additions and corrections to the paper.

Response to Reviewer #1:

The authors do some real and beautiful revisions. But I still think the manuscript haven't addressed the concerns on the lithium metal anodes. Like the reviewer 3 said, why do we need HNMG if lithium metal can do the same thing? In the response letter, the evidences for Li metal anodes seems to be very convincing, but there are actually not conformal with the content in manuscript. The manuscript stressed the outstanding rate capability and high areal capacity of HNMG in the abstract, but as to the comments on lithium metal anodes, especially in the response letter, they said their mass loading was 0.25 mg cm⁻² (In Figure 4B, the areal mass loading was 1-6 mg cm⁻², so the authors did their electrochemical performances tests under quite different conditions?), by the way, I think most readers will consider this ultra-low mass loading is not sufficient for providing high rate capability. And the electrolyte is different for lithium ion batteries and lithium metal anodes, the authors chose the 4M high concentrated ether-based electrolyte, which will have great improvement from normal carbonate-based electrolyte (for Li-ion batteries in manuscript). Although the other two reviewers suggested minor revisions, but I still think this manuscript is not suitable for nature communications.

Response:

We are very grateful for your comment. This comment was originally raised by the Reviewer #3. We have addressed this comment in the first revision, which was accepted by the Reviewer.

As known, thin films of lithium metal are often used as counter electrodes to quantify the performance of a new electrode material, which are often limited by the formation of dendrites, side reactions, and the inability to sustain a high current density. For the electrode with a mass loading of 0.25 mg/cm², the areal current density is ~ 0.2 mA/cm², considering the capacity of the HNMG at 1 C is around 744 mAh/g. For the study conducted at 60 C, the areal current density reaches ~ 12 mA/cm², which is far beyond that of a normal lithium-metal electrode can sustain. Testing the high-rate performance of electrodes with higher mass loadings therefore will be limited by the lithium-metal anodes. Consistently, a high-concentration ether-based electrolyte was used in this study (test at 60 C) simply to improve the plating/stripping stability, which will not affect the conclusion of high-rate performance.

Furthermore, in the first revision, we have demonstrated that the high-rate performance of the electrodes with high mass loading. For example, as shown in Figure 4B, the electrode with a mass loading of 6 mg cm⁻² provides a stable capacity of 642 mA h g⁻¹ at 2C rate for 500 cycles. As shown in Fig. 4C, at a high charge-discharge rate of 40 C, the HNMG anodes with a mass loading of 3 mg cm⁻² and 6 mg cm⁻² can still provide a capacity of 362 mA h g⁻¹ and 221 mA h g⁻¹, respectively. This work confirms the feasibility of fabricating high-mass-loading electrodes with high-energy-power performance.

Overall, the limitation of metal anodes is a well-known issue for the community. The main goal of this work is to develop high-performance graphene anodes, rather than the development of lithium-metal batteries. As suggested by other experts, we hope that you could appreciate the significance of this work and make a sound and fair judgment for this manuscript.

Response to Reviewer #2:

The authors had responded most of my concerns properly, some other minor suggestions before accepted: (1) The authors have added the content about the contribution of MgO as catalyst during the graphene growing, the references such as “ [27] Zhu, C. Y. & Akiyama, T. Cotton derived porous carbon via an MgO template method for high performance lithium ion battery anodes. Green Chem. 18, 2106-2114 (2016)” also mentioned the similar method to prepare LIB anode, the authors should show the different between them, for example, crystallinity, CVD method and so on.

Response: We sincerely appreciate the reviewer’s time and extremely valuable comments. As pointed, Zhu et al conducted an excellent work on the scalable fabrication of porous carbon by infiltrating MgO precursor within cotton followed by a carbonization process. Compared with a CVD-based approach, the performance of such carbons probably could be improved by removing the defects and amorphous moieties. We have incorporated the reference and related description in the revised manuscript.

Please see **page 11, line 14- line 18** in the revised manuscript:

“It is noteworthy that Zhu *et al*⁵⁵ demonstrated the fabrication of porous carbons for LIB anodes via carbonizing cotton infiltrated with a MgO precursor. Compared this low-cost and scalable method, the CVD approach could provide HNMG particles with less defect and amorphous carbon moieties, which could significantly improve the electrochemical performance.”

“55. Zhu, C. Y. & Akiyama, T. Cotton derived porous carbon via an MgO template method for high performance lithium ion battery anodes. *Green Chem.* **18**, 2106-2114 (2016).”

(2) The authors make lists to compare electrodes prepared via different template methods, the performance of this work shows the best. The authors said that the outstanding rate performance of the HNMG electrodes can be attributed to their effective charge transport. Thus, the authors should give more detailed explanations or references of the relationship between charge transport and defects, are the better performance just due to the reduction through microwave?

Response: Thank you for this important comment. We believe that defects in carbons should greatly affect the electrochemical performance. Considering carbon with sp^2 and sp^3 structure, the former structure could favor better electronic conductivity and more effective lithium-ion insertion and transport. In this context, microwave does play an essential factor, which

deserves a throughout investigation in the further. We have revised the manuscript accordingly.

Please see **page 11, line 18- line 25** in the revised manuscript:

“Various carbon materials have also been synthesized as anodes for LIBs using hard-template methods (Table S3). For carbon materials, a high content of sp^2 (less sp^3) moiety could improve the electronic conductivity and lead to more effective lithium-ion insertion and desertion. The high-quantity HNMG particles enable effective charge transport and provide outstanding mechanical and electrochemical robustness, endowing the electrodes with high areal mass loadings, high reversible capacity, superior rate capability and remarkable cycling stability. In this context, developing techniques (e.g., microwave irradiation) that can reduce the defect density could be of great interest towards high-performance carbons for LIBs.”

Reviewer #3:

Thank you for the additions and corrections to the paper.

Response: Thank you very much for your time and efforts!!!